# Concomitant Thoracic Spinal Hemangioma and Dural Arteriovenous Fistula: Case Report and Technical Note

**DOI:** 10.3390/reports8020074

**Published:** 2025-05-21

**Authors:** Stefano Vecchioni, Alessio Iacoangeli, Elia Giacomo Galli, Massimo Vissani, Alessandra Marini, Roberta Benigni, Michele Luzi, Roberto Trignani

**Affiliations:** 1Department of Neurosurgery, Azienda Ospedaliero Universitaria delle Marche, 60126 Ancona, Italy; stefano.vecchioni@ospedaliriuniti.marche.it (S.V.); eliagalli98@gmail.com (E.G.G.); massimo.vissani@ospedaliriuniti.marche.it (M.V.); alessandra_marini@ospedaliriuniti.marche.it (A.M.); roberta.benigni@ospedaliriuniti.marche.it (R.B.); michele.luzi@ospedaliriuniti.marche.it (M.L.); roberto.trignani@ospedaliriuniti.marche.it (R.T.); 2Department of Neurosurgery, Ospedale IRCSS San Martino, Università degli Studi di Genova, 16132 Genova, Italy

**Keywords:** hemangioma, dural arteriovenous fistula, spinal vascular malformation

## Abstract

**Background and Clinical Significance:** The coexistence of spinal hemangiomas and dural arteriovenous fistula (SDAVF) is uncommon. Unclear imaging and progressive neurological impairment require early surgical management. **Case Presentation**: A 76-year-old woman presented with progressive thoracolumbar pain and worsening bladder dysfunction. Magnetic resonance imaging (MRI) of the thoracic spine revealed a round-shape expansive lesion at T11 with spinal cord edema and homogeneous contrast enhancement. Despite a chronic presentation, the subacute progression of bladder dysfunction and spinal cord edema warranted timely intervention. Intraoperatively, a vascular malformation resembling a dural arteriovenous fistula (SDAVF), unrecognized at pre-operative imaging, was found in association, and histological examination confirmed the diagnosis of hemangioma. The mechanism of coexistence remains unclear, although venous hypertension due to fistula could induce vascular malformations. **Conclusions**: This case emphasizes the importance of thorough imaging, timely intervention and intraoperative assessment in patients presenting with a suspicion of spinal hemangioma; it may also provide awareness of potentially associated concurrent lesions such as SDAVFs, unrecognized at pre-operative imaging, and technical insights during surgery.

## 1. Introduction and Clinical Significance

Spinal hemangiomas are rare vascular lesions characterized by dilated, thin-walled capillaries with potential for hemorrhage and mass effect on the spinal cord, therefore necessitating accurate diagnosis and timely intervention [1,2]. Venous hypertension found in such settings is a proposed mechanism for the development of secondary vascular malformations, like arteriovenous malformations (AVMs) or dural arteriovenous fistulas (dAVFs), although such a co-occurrence is exceptionally rare. Magnetic resonance imaging (MRI) has become the cornerstone of diagnostic evaluation [3]; however, while spinal hemangiomas and SDAVFs each have distinct imaging characteristics, typical MRI findings for SDAVF are not always evident. These lesions can lead to significant neurological morbidity, with clinical presentations ranging from asymptomatic cases to severe neurological manifestations such as pain, sensory disturbances, motor weakness, and bladder or bowel dysfunction [1]. Overlap in their clinical presentation—particularly with progressive myelopathy and cord edema—can furtherly obscure diagnosis. The role of additional imaging modalities, such as spinal angiography, in identifying associated vascular anomalies or concurrent pathologies remains an area of ongoing investigation [4].

This case report presents information on an elderly woman with a spinal intradural hemangioma and a concurrent spinal dural arteriovenous fistula, a rare and diagnostically challenging combination. By reviewing the current literature on spinal hemangiomas and concurrent vascular malformations, we aim to highlight the unique aspects of these rare conditions, the diagnostic challenges they pose, and the importance of surgical insight when advanced imaging techniques are not feasible or readily available to ensure accurate diagnosis and effective treatment. The case underscores the risk of insufficient care if concurrent lesions are overlooked and emphasizes the need for a high index of suspicion in patients with complex spinal vascular pathologies.

## 2. Case Presentation

A 76-year-old woman presented with a six-month history of progressive dorso-lumbar pain. Neurological examination revealed significant pain but preserved lower limb strength; bladder dysfunction was present and worsening overtime with a sub-acute trend in the hours before surgery. Magnetic resonance imaging (MRI) of the thoracic spine revealed an expansive lesion at the T11 vertebral level [Figure 1]. Preoperative spinal angiography was not performed, as the imaging findings were interpreted as consistent with a hemangioma without clear evidence of flow voids or abnormal vascular connections. The patient’s medical history included long QT syndrome, osteoporosis, and bilateral carpal tunnel syndrome. Neurosurgical evaluation indicated the need for timely surgical resection due to the ongoing worsening of symptoms and to the unclear imaging.

### 2.1. Surgical Technique

The patient was positioned prone, and a midline incision was performed, followed by bilateral muscle dissection and laminoplasty at T11–T12 using a piezoelectric osteotome (Mectron s.p.a—Carasco, Italy) [Figure 2]. Under the operating microscope, a dural opening was made, exposing the lesion, which appeared as an exophytic protrusion from the surface of the left posterior spinal cord, surrounded by a vascular malformation resembling a dural arteriovenous fistula (SDAVF) with multiple arteriovenous malformations on the pial surface [Figure 3]. Coagulation and disconnection of the fistulas were performed first by using bipolar cautery and scissor separation [Figure 3]; a detailed image of the hemangioma and the associated fistula is shown [Figure 4]. An incision was made around the lesion, which exhibited firm consistency. Due to the inability to remove the pathological tissue without traction, an ultrasonic aspirator was used for the microsurgical resection. Meticulous hemostasis was achieved. The dura mater was sutured, and a laminoplasty was performed using plates and screws. Standard wound closure was performed.

### 2.2. Postoperative Course

Postoperatively, the patient exhibited mild lower limb weakness (MRC 4), persistent paresthesia and hypoesthesia in the left leg, and significant postvoid residual volumes requiring intermittent catheterization. A rehabilitation consultation recommended an extensive homebased rehabilitation program, and the patient fully recovered with no neurological deficits after 2 months.

Histopathological examination later confirmed the diagnosis of hemangioma, with immunohistochemical staining positive for ERG and CD34.

A contrast-enhanced MRI was performed at 3 months, demonstrating the absence of residual pathology [Figure 5].

## 3. Discussion

Spinal hemangiomas are typically benign, incidental findings, while spinal vascular malformations can lead to significant neurological deficits if left untreated. In more detail, spinal AVMs and dAVFs are abnormal vascular connections that can lead to venous hypertension, spinal cord ischemia, or hemorrhage, often presenting with progressive neurological deficits [5,6]. The distinct clinical presentations and pathophysiological mechanisms of these lesions underscore the importance of a comprehensive diagnostic approach, particularly in cases where concurrent pathologies may be initially overlooked in imaging. The co-occurrence of spinal hemangiomas with other vascular malformations is exceptionally rare, with only one documented case reported in the literature [5], where initial imaging findings were subtle, and the concurrent dAVF was only identified through detailed angiographic evaluation. The mechanism of vascular malformation generation remains unclear; nevertheless, in this case, we hypothesize a causal link with preexisting vascular lesions, since the proximity of the fistula and the lesion hinted to hemodynamic interplay. A clear connection between the fistula and the lesion was evident intraoperatively, confirming this thin vessel as a possible blood supply for the hemangioma, owing to the location, morphology, and behavior upon microsurgical manipulation. According to Xu W Ren J et al., a dural AVF may precede and contribute to the development of vascular malformations by inducing intramedullary venous hypertension, as in the pathophysiology of intracranial cavernous malformations combined with developmental venous anomalies (DVA), resulting in a cascade of pathophysiological signaling events leading to vascular disorders [7]. Animal studies by Zhu et al. and Li et al. demonstrate that increased venous pressure leads to the upregulation of hypoxia-inducible factor-1 (HIF-1) and subsequent vascular endothelial growth factor (VEGF) expression, initiating within a day and peaking within a week of hypertension onset [8,9,10,11,12,13,14]. This VEGF surge promotes abnormal angiogenesis and vascular remodeling, as shown by the elevated microvessel density and dural AVF formation in hypertensive animals [14]. Furthermore, Chen et al. observed the simultaneous upregulation of matrix metalloproteinase-9, indicating a broader, sustained angiogenic response [15]. These findings are corroborated by human studies as follows: Uranishi et al. reported robust VEGF and basic fibroblast growth factor (bFGF) expression in dural AVF tissue, particularly in the context of venous outflow obstruction [16]. Clinically, Lawton et al. and Hetts et al. have shown that the degree of venous engorgement correlates with angiogenic activity and neurological deterioration, respectively [17,18]. While most data are derived from cranial models, spinal studies (e.g., Hassler, Hetts) confirm that venous congestion similarly disrupts cord perfusion and function, suggesting that the pathogenic sequence—venous hypertension, molecular signaling, angiogenesis, and malformation—may operate across both cranial and spinal compartments [17,18,19,20]. However, the unclear pathophysiology of this combination, in addition to its rarity, poses significant diagnostic and therapeutic challenges, as the presence of concurrent lesions may complicate clinical presentation, imaging interpretation, and treatment planning.

The identification of concurrent vascular lesions requires a high index of suspicion and meticulous imaging analysis, which may not always be feasible in common clinical practice due to worsening of symptoms and need for timely intervention.

The diagnostic evaluation of spinal vascular malformations relies heavily on advanced imaging techniques. MRI is the primary screening tool due to its superior soft tissue contrast, highlighting characteristic features such as a “popcorn” appearance, mixed T1 and T2 signal intensities, and a low-signal rim on T2-weighted images in hemangiomas; meanwhile, spinal SDAVF usually show peculiar features as serpiginous flow voids and T2 hyperintensity, warranting further testing. As SDAVF are not always readily recognizable on MRI, spinal angiography remains the gold standard for the definitive characterization of vascular anatomy and lesion type through invasive testing. In this case, the concurrent presence of a hemangioma and SDAVF posed a diagnostic challenge, necessitating a multimodal imaging approach [8].

In retrospect, the intraoperative identification of an arteriovenous structure suggests that preoperative DSA could have clarified the preoperative diagnosis, possibly leading to adjustments in the therapeutic plan, e.g., preoperative endovascular embolizations of the SDAVF. Treatment strategies for spinal hemangiomas and concurrent vascular malformations are primarily guided by the patient’s clinical presentation and the lesion’s characteristics. Surgical resection remains the mainstay of treatment for spinal hemangiomas and vascular malformations; it is generally recommended for symptomatic cases, particularly those with evidence of hemorrhage or progressive neurological decline [9]. Microsurgical techniques, including laminotomy, have been shown to yield favorable outcomes, with studies reporting neurological improvement in 41–80% of patients [10]. In cases of concurrent lesions, both lesions may need to be addressed surgically in the same operation to achieve optimal outcomes [5]. In this case, we preferred performing a laminoplasty to preserve stability, and a piezoelectric osteotome was employed to maximize safe bone removal and minimize the risk of soft tissue damaging and dural tear [11,12]. Conservative management, including observation and serial imaging, is typically reserved for asymptomatic or non-hemorrhagic lesions, although the long-term outcomes of this approach are still not sufficiently defined [9].

## 4. Conclusions

In patients presenting with suspected spinal hemangiomas on imaging and progressive neurological symptoms, the coexistence of spinal hemangiomas and SDAVF is a rare event, which requires thorough clinical and surgical attention in order to avoid missing additional pathologies that could possibly contribute to the clinical outcome. As the physiopathology of this rare combination is not clearly understood, further investigation may provide reliable clinical and radiologic markers to prompt advanced preoperative testing.

At present, a multimodal approach combining adequate preoperative imaging, microsurgical resection, and postoperative rehabilitation is crucial for the optimization of clinical outcomes.

## Figures and Tables

**Figure 1 reports-08-00074-f001:**
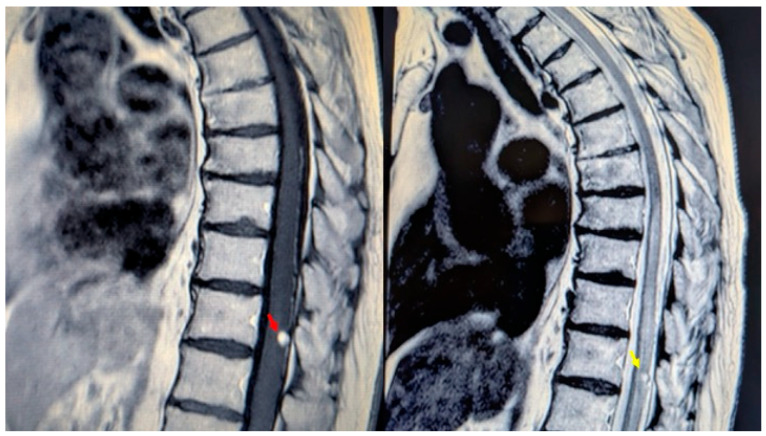
MRI of the thoracic spine. Left: T1-weighted image showing the expansive lesion at the T11 level (red arrow). Right: T2-weighted image revealing the lesion with surrounding edema (yellow arrow).

**Figure 2 reports-08-00074-f002:**
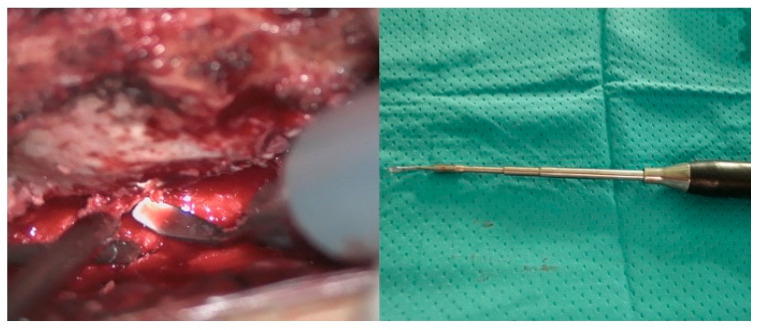
Piezoelectric osteotome while performing laminoplasty (**left**) and intra-operative view of a long tip used to reach deep surgical fields.

**Figure 3 reports-08-00074-f003:**
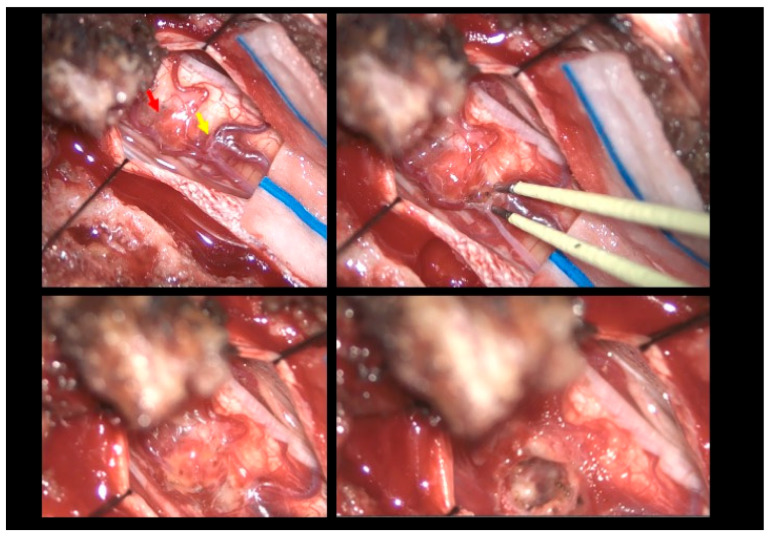
Intraoperative findings. **Upper left**: Visualization of the hemangioma (red arrow) associated with the vascular malformation resembling a dural arteriovenous fistula (yellow arrow). **Upper right**: Sealing of the fistula using bipolar cautery. **Bottom left**: Macroscopic color change following fistula closure. **Bottom right**: Macroscopic appearance following hemangioma removal.

**Figure 4 reports-08-00074-f004:**
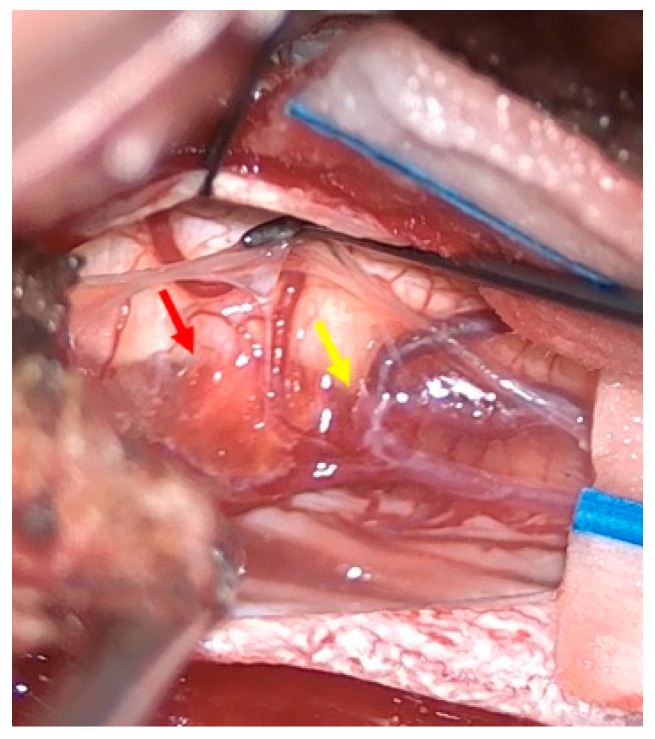
Intraoperative findings. The hemangioma (red arrow) and arteriovenous fistula (yellow arrow).

**Figure 5 reports-08-00074-f005:**
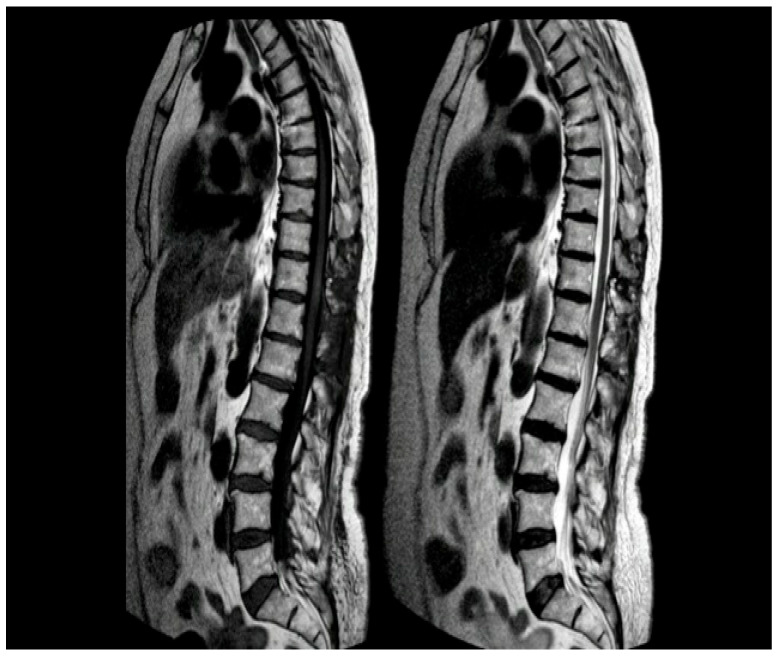
Post-operative MRI of the thoracic spine at 3 months. **Left**: T1-weighted image showing the absence of hyperintense findings. **Right**: T2-weighted image showing edema resolution.

## Data Availability

Data are unavailable due to privacy and ethical restrictions.

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
