# Peer review of "Concomitant Thoracic Spinal Hemangioma and Dural Arteriovenous Fistula: Case Report and Technical Note"

_reports, 2025, doi:10.3390/reports8020074_

Round 1

Reviewer 1 Report

Comments and Suggestions for Authors

The author reports a rare case of spinal hemangiomas combined with dural arteriovenous fistula.  Generally, It is difficult to detect the small DAVF in the MRI before the surgery. this case provides awareness of spinal hemangiomas potentially associated with SDAVF and technical in the surgery. My advice is to revise the "D11" to "T11 "and add the pathology picture of the lesion.

Author Response

Reviewer #1: The author reports a rare case of spinal hemangiomas combined with dural arteriovenous fistula.  Generally, It is difficult to detect the small DAVF in the MRI before the surgery. this case provides awareness of spinal hemangiomas potentially associated with SDAVF and technical in the surgery. My advice is to revise the "D11" to "T11 "and add the pathology picture of the lesion.

Response: Thank you for appreciating our work and for the esteem addressed to our manuscript; manuscript has been changed through all the text with your important suggestion. Adding the pathology picture would enrich the value of the manuscript but unfortunately we are not able to publish the pathologic image due to Pathology Lab and Institutions restrictions. Thank you again for taking time to consider our article.

Reviewer 2 Report

Comments and Suggestions for Authors

This is an interesting and well-documented case report describing the rare coexistence of a thoracic spinal hemangioma and a spinal dural arteriovenous fistula (SDAVF). The authors present the case clearly, with adequate imaging, surgical details, and a relevant review of the literature. The inclusion of intraoperative and postoperative imaging, as well as histological confirmation, strengthens the diagnostic value of the report.

  • Diagnostic Work-up and Imaging Discussion: While the authors highlight the diagnostic challenge of this coexistence, the preoperative imaging strategy could be described in greater detail. For example, it would be valuable to know whether spinal angiography was considered or performed preoperatively, as well as the rational behind the choice. Since the SDAVF was only found intraoperatively, the case might benefit from a short discussion on how preoperative DSA could have changed the management or influenced the differential diagnosis.
  • The section describing the postoperative course is currently placed before the “Surgical Technique” section, which disrupts the chronological flow of the case presentation. It would improve the clarity and logical structure of the manuscript to move the postoperative outcomes to the end of the surgical section or present them as a dedicated “Postoperative Course” subsection immediately after surgery is described.
  • Mechanistic Hypothesis: The discussion touches on the potential pathophysiological mechanism of venous hypertension promoting hemangioma formation. While this is plausible, the argument would be stronger if supported by more precise references or with clearer differentiation between hypothesis and established findings. The paragraph could benefit from rephrasing for clarity and scientific tone.

Author Response

Thank you for appreciating our work and for the esteem addressed to our manuscript; manuscript has been changed through all the text with your important suggestion.

Clinical onset and worsening of symptoms as well as unclear MRI findings did not allow for pre-operative DSA although it could have been useful to anticipate intraoperative findings. We revised discussion in order to clarify literature basis on which our pathophysiological hypothesis finds strength. Paragraphs have been changed of order to better clarify the case management.

Reviewer 3 Report

Comments and Suggestions for Authors

In their article, the authors discuss a case with a spinal cavernoma and possible dural AV fistula.
The case is clearly presented, but some things need to be revised.

Abstract:
-Medical history of 6 months does not explain urgent surgery

Full text:
- Please pay attention to a logical sequence / red thread. The case is briefly alluded to several times before it is explicitly explained. However, the clinical pictures / diagnostics should first be discussed in detail in the introduction

Case description:
- Typical features of a dAVF on MRI are missing
- Oedema in a cavernoma may also be due to haemorrhage
- Intraoperatively, a vessel can be recognised on the mylon surface. Where does the suspicion of a dural fistula come from? Where is the fistula point? Typically, this must be treated at the transition to the intradural. The lack of aniography must be critically analysed
- Reference to Figure 3 does not match the previous text
- A laminotomy was not performed but a laminoplasty

Discussion:
- Note the spelling of sources (et al. - dot at the end)

- The presence of a dural fistula is very hypothetical and cannot be confirmed on the basis of the MRI and the intraoperative findings. A blocked vein / DVA would be conceivable?
- There is also no recognisable common thread in the discussion. Please discuss the clinical picture on the basis of the literature. Case description and literature are a little mixed up
- What do laminectomy and posterior approaches mean? (Laminectomy is a dorsal approach and ventral approaches are unusual)
- Some phrases are repeated / are redudant: e.g. line 130 ... highlights the diagnostic complexities...; line 133: underscoring the importance thorough diagnostic workup...

Author Response

Thank you for appreciating our work and for pointing these observations out. We agree with your significant and inspiring comment. Therefore, we have revised the manuscript trying our best to provide the necessary clarifications and here is a point-by-point explanation of the changes made.

  • Abstract: timely surgery was justified by sub-acute worsening of bladder function
  • Introduction was modified in order to improve the logical sequence of the description and to provide a first discussion of the case
  • Missing typical features such as clear evidence of flow voids or abnormal vascular connections was underscored in the case description
  • We critically analysed the lack of preoperative angiography which was missing due to missing preoperative MRI findings suspicious for fistula and to the need for timely intervention. We improved figures to better indicate where we suspected the fistula point
  • Reference to figure 3 was corrected as well as laminoplasty instead of laminotomy
  • Et al. spelling was corrected
  • Discussion was rewritten trying to clarify which was the evidence of MRI and intraoperative findings, the literature basis on which the manuscript finds strength and the reason why an angiography was not feasible
  • Posterior approaches were corrected including only a sentence on laminoplasty
  • Redundant phrases were cancelled

Round 2

Reviewer 3 Report

Comments and Suggestions for Authors

  • there is duplication in the case description (from line 89)
  •  Although described differently, the illustrations have not been optimised. The presence of a spinal dural AV fistula cannot be verified here.
  • Regarding Figure 4, the reference in the text still does not fit: Good neurological recovery described in the text followed by reference to Figure 4, which shows an unremarkable MRI
  • Please add that an MRI was performed (see Figure 4) - The clinical condition does not necessarily correlate with the MRI imaging
  • A critical discussion regarding a DVA (as often seen intracranial) was already recommended beforehand

Author Response

Thank you for appreciating our work and for pointing these observations out. We agree with your significant and inspiring comment. Therefore, we have revised the manuscript trying our best to provide the necessary clarifications and here is a point-by-point explanation of the changes made.  

  • duplications have been removed;
  • spinal fistula was better described adding a picture showing intraoperative findings in detail
  • reference of the post operative MRI was fixed along with adding the post operative clinical condition;
  • figure 4 is now renamed as figure 5
  • discussion was critically re-written